# Efficacy of antimalarial drugs for treatment of uncomplicated *falciparum* malaria in Asian region: A network meta-analysis

Cho Naing[1,2]☯ *, Maxine A. Whittaker[2]☯, Norah Htet Htet[1], Saint Nway Aye[1], Joon Wah Mak[1]

**1** International Medical University, Kuala Lumpur, Malaysia, **2** Faculty of Tropical Heath and Medicine, James Cook University, Queensland, Australia

☯ These authors contributed equally to this work.
* cho3699@gmail.com

## Abstract

### Background

The WHO recommends artemisinin-based combination therapies (ACTs) for the treatment of uncomplicated *falciparum* malaria. Hence, monitoring the efficacy of antimalarial drugs is a key component of malaria control and elimination. The published randomized trials that assessed comparisons of ACTs for treating uncomplicated *falciparum* malaria reported conflicting results in treatment efficacy. A network meta-analysis is an extension of pairwise meta-analysis that can synthesize evidence simultaneously from both direct and indirect treatment comparisons. The objective was to synthesize evidence on the comparative efficacy of antimalarial drugs for treatment of uncomplicated *falciparum* malaria in Asian region.

### Methods

Relevant randomized trials that assessed efficacy of antimalarial drugs for patients having uncomplicated *falciparum* malaria in Asian region were searched in health-related databases. We evaluated the methodological quality of the included studies with the Cochrane risk of bias tool. Main outcome was treatment success at day 28 as determined by the absence of parasiteamia. We performed network meta-analysis of the interventions in the trials, and assessed the overall quality of evidence using the GRADE approach.

### Results

Seventeen randomized trials (n = 5043) were included in this network meta-analysis study. A network geometry was formed with 14 antimalarial treatment options such as artemether-lumefantrine (AL), artemisinin-piperaquine, artesunate-amodiaquine, artesunate-mefloquine (ASMQ), artesunate-chloroquine, artesunate-mefloquine home treatment, artesunate-mefloquine 2-day course, artesunate plus sulfadoxine-pyrimethamine, chloroquine, dihydroartemisinin-piperaquine (DHP), dihydroartemisinin-piperaquine home treatment, dihydroartemisinin-piperaquine 4-day course, dihydroartemisinin-piperaquine and added

**Funding:** The authors received no specific funding for this work.

**Competing interests:** The authors have declared that no competing interests exist.

artesunate, sulfadoxine-pyrimethamine. A maximum number of trials included was DHP compared to ASMQ (n = 5). In general, DHP had better efficacy than AL at day 28 (DHP vs AL: OR 2.5, 95%CI:1.08–5.8). There is low certainty evidence due to limited number of studies and small trials.

## Discussion/ Conclusions

The findings suggest the superiority of DHP (3–day course) to AL and other comparator ACTs are with the overall low/very low quality of evidence judgements. Moreover, one drug regimen is better than another is only if current drug-resistance patterns are at play. For example, the AL might be better than DHP in areas where both artemisinin and piperaquine resistance patterns are prevalent. For substantiation, well-designed larger trials from endemic countries are needed. In the light of benefit versus harm concept, future analysis with safety information is recommended.

## Introduction

Malaria caused by *Plasmodium falciparum* is responsible for >90% of malaria cases and almost all of the malaria deaths worldwide. According to the WHO report, there was no significant progress in reducing global malaria during the period 2015–2017 [1]. The Global technical strategy (GTS) 2020 milestones include the elimination of malaria in at least 10 countries that were malaria endemic in 2015 [2]. In fact, malaria is a preventable and curable disease. The WHO recommends artemisinin-based combination therapies (ACTs) for the treatment of uncomplicated malaria caused by *P. falciparum*. The primary advantage of the combination therapy is that the artemisinin quickly and drastically reduces the majority of malaria parasites, and the partner drug clears remaining small number of parasites [3,4]. Thus far, ACTs is the newest class of antimalarials that are used worldwide including in the Greater Mekong Sub-region (GMS). There are reports on evidence of artemisinin resistant hotspots in Cambodia, Thailand and on the Thai-Myanmar border [1,2]. The emergence of *falciparum* resistance to artemisinins would not only limit treatment options in the affected areas, but could also compromise the management of uncomplicated malaria cases in other areas where ACT is widely recommended [5]. Containment of parasites developing resistance to anti-malarial drugs is one of the major goals to progress from malaria control towards elimination [4,5]. Hence, monitoring the efficacy of antimalarial drugs is a key component of malaria control and subsequent elimination.

Five ACTs recommended by WHO for treatment of uncomplicated *P. falciparum* malaria are: artemether—lumefantrine (AL), artesunate- amodiaquine (ASAQ), artesunate- mefloquine (ASMQ), artesunate plus sulfadoxine-pyrimethamine (ASSP) and dihydroartemisinin-piperaquine (DHP) [3,5]. Protecting the efficacy of ACTs as the current first- and second-line treatment for *P.falciparum* malaria is one of a top global public health priority [1,3]. However, the question is which antimalarial drugs offers the greatest benefits (efficacy) for treatment of uncomplicated *P. falciparum* malaria at day 28? There are published Cochrane systematic reviews [6,7] and non-Cochrane systematic reviews/meta-analyses [8,9], assessing head-to-head comparisons of ACTs for treating uncomplicated *P.falciparum* malaria. These reviews reported conflicting results. For example, a review by Zani and associates reported that in Africa, there was better efficacy in DHP than AL at day 28. However, such relationship was not

shown in Asia [7]. This reflects that efficacy of antimalarial is related to whether the area is with artemisinin-sensitive parasite populations or not [7,9]. Hence, we performed a network meta-analysis (NMA) that can synthesize evidence simultaneously from both direct and indirect treatment comparisons [10]. For instance, even when no head-to-head trial is available, studies evaluating A versus B and B versus C can be used to compare A and C indirectly through the NMA approach. Indirect comparisons must be connected by at least one common comparator (i.e. treatment B in this example). An assumption required for the NMA is 'transitivity' that trials should be comparable in all characteristics [11]. On the whole, the objective of present study was to synthesize evidence on the comparative efficacy of currently used antimalarial drugs for treatment of uncomplicated *falciparum* malaria in Asian region.

## Materials and methods

The current study adhered to the preferred reporting items for network meta-analyses (PRISMA-NMA) [12] (S1 Table).

### Search strategy

We searched relevant studies in the health-related electronic database including MEDLINE, Medline-in-Process, OLD Medline, EMBASE, and the Cochrane library using the texts including "malaria" "Artemisinin-based combination therapy", "randomized trial" "humans" with Boolean operators. The search was done according to guidance provided in the Cochrane Handbook for Systematic Reviews of Interventions [13] and in consultation with an information specialist. The search strategies for the MeSH terms are listed in S2 Table. Additionally, we searched in, ClinicalTrials.gov, EU Clinical Trials Register and WHO International Clinical Trials Registry Platform for ongoing trials. Search was limited to studies published in English language until June 2019.

### Selection of study

Eligible studies were identified through the PICOS format [13].

Study Population (P): Participants having uncomplicated *falciparum* malaria residing in Asian region, regardless of gender and age were included.

Uncomplicated malaria caused by the *P. falciparum* parasite in this study is defined as patients having symptoms that are non-specific in the presence of *P. falciparum*, but in the absence of clinical or laboratory findings of severe organ dysfunction.

An operational definition of the Asian region for this particular study covers countries in three regions of Southeast Asia, South Asia and East Asia.

Interventions (I): Anti-malarial drugs for the treatment of uncomplicated *falciparum* malaria were considered. Different dosages of an antimalarial regimen were considered as individual treatments.

Comparisons (C): An alternative antimalarial drug or placebo were included.

Study Outcomes (O): Main outcome was cure rate at day 28 (defined as the proportion of patients with clearance of asexual *P. falciparum* parasitaemia within seven days of initiation of trial drug, without subsequent recrudescence within 28 days after initiation of study). Recrudescence was defined as the existence of positive blood smears after initial clearance of parasites from the peripheral blood [14].

Study design (S): Randomised clinical trials (RCT), conducted in Asian region.

Studies were excluded, if they did not meet the inclusion criteria. Studies on pregnant women and travellers were not considered.

## Data extraction and management

One investigator (SNA) screened title and abstracts on the basis of RCTs that assessed human *falciparum* malaria. The same investigator extracted information from the RCTs included. Information collected were study characteristics, intervention and comparators and outcomes. Information collected were cross-checked by another investigator (CN). Any discrepancies were settled by discussion.

## Methodological quality assessment

The methodological quality of the RCTs was evaluated using the Cochrane risk of bias tool [13]. Three domains (adequate sequence generation, allocation concealment and blinding of participants and outcome assessors) were assessed for the risk of bias assessment for each trial. The ratings were noted (i.e. high risk, unclear risk, low risk) for the risk of bias category in subsequent Grading of Recommendations Assessment, Development, and Evaluation (GRADE) assessment. [15].

## Data synthesis

Main outcome in this review was treatment success (cure rate) by anti-malarial treatment at day 28. We preferred the intention-to-treat (ITT) analysis over another analysis, whenever it was available. We performed pair-wise meta-analyses of all available within-study comparisons, followed by subsequent network meta-analyses.

**Pairwise comparison**: When the studies reported in similar ways, we did head-to-head comparisons as a direct pairwise meta-analyses. An odds ratio (OR) and its 95%confidence interval (CI) were computed for the dichotomous variables. Between-study heterogeneity was assessed with the $I^2$ statistic. We pooled ORs with a DerSimonian-Laird random-effects model in the presence of substantial heterogeneity ($I^2 > 50\%$). Publication bias was investigated with the contoured-enhanced funnel plot [13].

**Network meta-analyses**: We performed NMA within a frequentist framework using random-effects models [16–18]. We established network mapping. An assumption of NMA is 'transitivity' that the trials comparing different sets of interventions should be similar enough in their characteristics [11]. We also investigated another assumption of NMA such as network 'inconsistency' (i.e. disagreement between the different sources of evidence) with the use of the global Wald test for inconsistency [19,20]. We also checked, if there were concerns with 'intransivity' [21]. For a ranking of the effectiveness, we reported 'Surface Under the Cumulative Ranking Curve' (SUCRA) [11, 18]. SUCRA = 1 or 0 was indicated the rank of an intervention drug as first or last, respectively. Statistical significance was set at $p$ value $\leq 0.05$.

## Assessing the quality of evidence

We assessed the quality of evidence derived from the pairwise and NMA, following the GRADE approach, as described elsewhere [15,17,21,22]. For direct comparison, we rated evidence on the five categories; study limitations (risk of bias), precision, consistency of results, directness of evidence and publication bias, using the standard GRADE approach. We then evaluated the overall confidence in estimates of effect for treatment efficacy for each direct comparison as 'high', 'moderate', 'low' or 'very low' quality of evidence. For indirect comparison, we rated evidence from the most dominant first-order loop by first taking the lowest certainty of direct comparisons. We did not rate on intransitivity [21] in the absence of important imbalance in the distribution of effect modifiers (e.g. age, gender) across included trials. For NMA mixed estimates, we started with the higher quality of the two certainty ratings and rated down certainty for incoherence (degree of

inconsistency between direct and indirect effect estimates) in the final quality rating [21,22]. Data analysis was employed with STATA 15.0 (StataCorp, TX).

## Results

### Trials included

Fig 1 illustrates the study selection process. The initial search produced 13640 hits. After removal of duplicates and screening of titles and abstracts, 34 full-text papers were evaluated and 17 studies, incorporating 5043 total number of patients were finally selected for this review [14, 23–38]. The highest number of trials include DHP compared to ASMQ (n = 5), followed by artemisinin-piperaquine (AMPQ) to DHP (n = 3) and AL to ASMQ (n = 2). A summary of the 17 excluded studies is provided in S3 Table.

Table 1 provides the key characteristics of the studies identified. The number of participants varied from 47 [14] to 769 [36]. The majority of participants in the studies included were males (range from 51% to 96%) with mean age between 5.9 to 29 years. The distribution of studies is presented in S4 Table. Three studies were three-arm RCTs [23,24,30] and one study was a four-arm RCT [29] and the remaining 13 studies were two-arm RCT. Studies were conducted in 8 countries such as Cambodia, India, Indonesia, Laos, Myanmar, Nepal, Thailand and Vietnam. One multi-country study was conducted in India, Laos and Thailand [36].

The number of studies with unclear risk of bias in sequence generation (53%) and that with high risk of bias in the blinding status was 58.8%. Overall risk of bias assessment revealed that most studies had an unclear/high risk of bias due to insufficient information on allocation concealment and the blinding status of the RCTs (S5 Table).

### Fourteen-node analysis

Fig 2 shows a network plot of treatment success with 14 antimalarial treatment options. These options of antimalarial regimens included AL, AMPQ, ASAQ, artesunate plus chloroquine (ASCQ), ASMQ, artesunate-mefloquine home treatment/not supervised (ASMQh), artesunate-mefloquine 2-day course (ASMQ2), ASSP, chloroquine (CQ), DHP, dihydroartemisinin-piperaquine home treatment/not supervised (DHPh), dihydroartemisinin-piperaquine 4-day course (DHP4), dihydroartemisinin-piperaquine and added artesunate (DHPAS), sulfadoxine-pyrimethamine (SP). As seen in the present network map, none of the studies compared ASSP and CQ directly, but each had been compared with a common comparator AL. We may assume an indirect comparison of ASSP and CQ on the direct comparison of ASSP and AL and the direct comparison of CQ and AL. Fig 3 presents forest plot with effects for each study, estimates from direct pairwise meta-analysis and mixed estimate from the network meta-analysis.

Pairwise-analysis of the relative efficacy of antimalarial drugs for treating uncomplicated *P. falciparum* malaria was reported that there were comparable cured rates between the treatment regimens, spanning both benefit and harm, except one comparison (i.e. AL versus CQ) (S1 Fig). For instance, DHP compared to AL in a single trial and there was a superiority of DHP in cure rate at day 28 (OR 2.5, 95%CI: 1.08 to 5.8) (Table 2). The results of the network meta-analysis are presented in Fig 4. In general, DHP was better than many comparators in terms of efficacy at day 28. For instance, DHP was superior to ASCQ (OR: 11.21,95% CI 3.4–36.89). Of note is that there was small number of studies in many comparisons and 95%CIs were (very) wide. For instance, DHP versus AL was done in a single study.

The global Wald test showed the presence of consistency in the network [Chi$^2$ (4) = 3.02, p = 0.8089] (Fig 3). Tests of local incoherence did not show any inconsistent loops for efficacy at day 28 (S2 Fig). The comparison adjusted funnel plots of the network meta-analysis [39] for efficacy at day 28 was not suggestive for publication bias (S3 Fig).

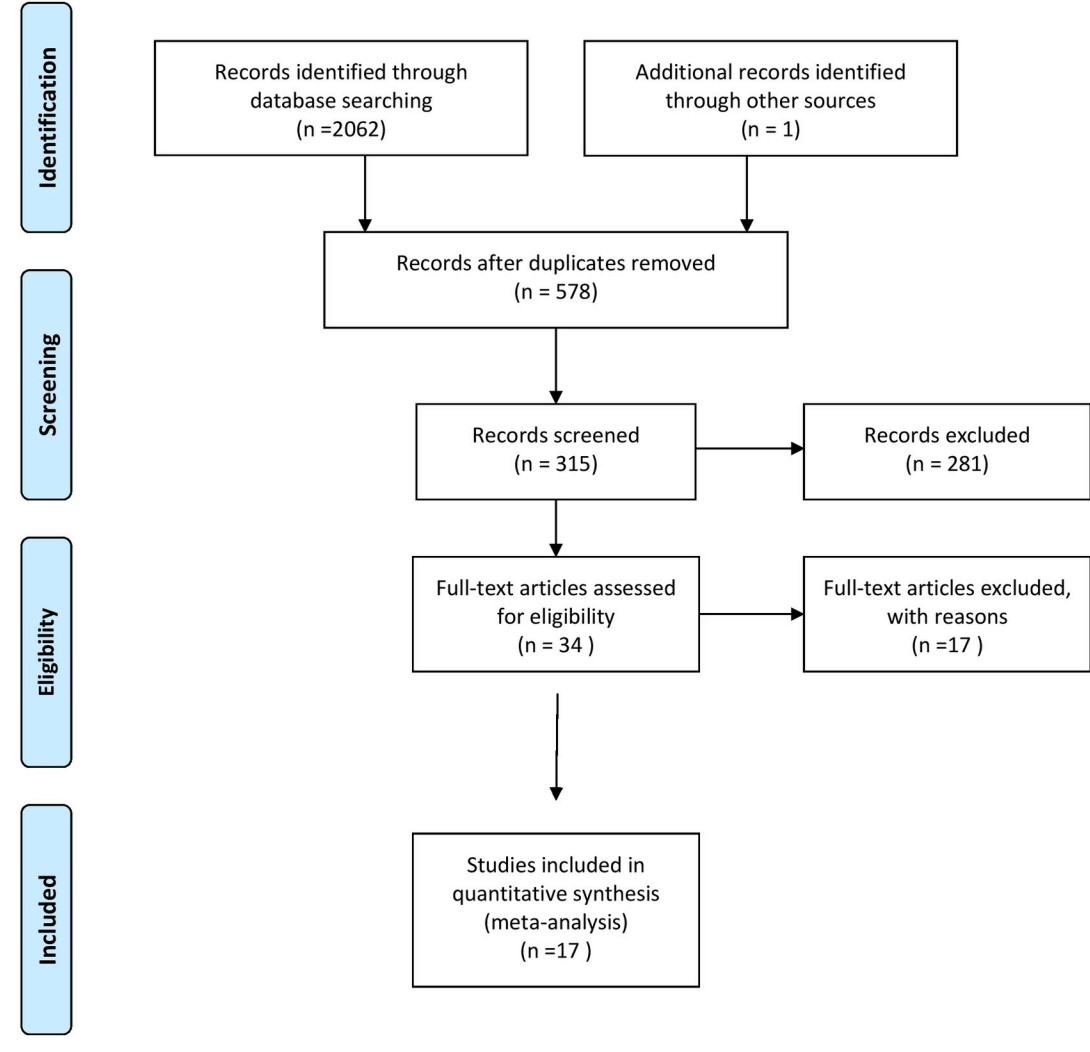

**Fig 1. Study selection process.**

### Treatment relative ranking

Treatment-relative ranking in this network meta-analysis is presented in (S6 Table). DHP had the highest probability of being the best choice for treating patients with uncomplicated *P. falciparum* (S4 Fig). The SUCRA and ranking results are subjected to the small number of participants in some studies and wide estimates spanning from benefit to harm. As the evidence on which the SUCRA rankings are of very low quality, they are untrustworthy [40]. Predictivity intervals of mixed estimates are presented in Fig 5. Although their confidence intervals suggest an association, the respective predictive interval crosses the line of no effect and suggests that future studies might favour either treatment. We therefore made overall evidence through the GRADE approach rather than the SUCRA rankings (Table 2).

Overall, we observed a low certainty evidence whether any antimalarial regimens included in this study were better in clearance of parasitemia at day 28 since the certainty of the evidence was assessed as low.

**Table 1. Characteristics of the studies included in the network meta-analysis.**

| | Author, Year of publication [ref.no] | Study period | Country | Setting | Study arms | Interventions | Participants | Male | Age, mean in year (±SD) or median and range | PCR | Under supervision | Funding |
|---|---|---|---|---|---|---|---|---|---|---|---|---|
| 1 | Rachmawati, 2010 [14] | 1/2009 - 7/2009 | Indonesia. | H (in-pt) | 2 | AL, ASSP | 47 | 59.5% | 5.9 ±3.46 | No | Yes | Not reported |
| 2 | Ashley, 2004 [23] | 7/2002–4/2003 | Thailand | H (in-pt) | 3 | ASMQ, DHP, DHPAS | 731 | 78.1% | Adults (25.3± 8.2) | No | Yes | Holleykin Pharmaceutical |
| 3 | Ashley, 2005 [24] | 4/2003–4/2004 | Thailand | OPD | 3 | ASMQ, DHP, DHP4 | 499 | 60.5% | Any age (21, 3–57) | Yes | Yes | MMV; Wellcome-Mahidol; Wellcome Great Britain. |
| 4 | Kshirsaga, 2000 [25] | 6/1996–1/1997 | India | H (in-pt) | 2 | AL, CQ | 179 | 96% | 29 (17–66) | Yes | Yes | |
| 5 | Lefevre, 2001 [26] | 9/1998-1/1999 | Thailand | H | 2 | AL, ASMQ | 219 | 70% | 50 (12–71) | Yes | Yes | Novartis Pharma AG. |
| 6 | Huong, 2003 [27] | NA | Vietnam | H (in-pt) | 2 | ASSP, ASCQ | 123 | 51% | Any age, 10.3 ± 11.3 (4–65) | Yes | Yes | male% & age in the ASSP gr |
| 7 | Silachamroon, 2005 [28] | NA | Thailand | H (in-pt) | 2 | ASMQ, ASMQ2 | 120 | 70.8% | Adults (25.6± 10.1) | No | Yes | WHO/RBM/ Mahidol University |
| 8 | Smithuis, 2006 [29] | 11/2003 - 4/2004 | Myanmar | OPD | 4 | ASMQ, ASMQh, DHP, DHPh | 652 | 52% | 3 age-gr; 58.2%(5–14 yr) | Yes | Yes (Gr1 No (Gr 2) | MSF (Holland); |
| 9 | Song, 2011 [30] | 7/2005 - 10/2005 | Cambodia | H (in-pt) | 3 | AL, AMPQ,DHP | 220 | 73% | 3 age-gr; 80% (>15 yr). | Yes | Yes | Science & Technology Planning Project, MOST/China |
| 10 | Thanh, 2009 [31] | 9/2006 - 12/2007. | Vietnam | Health station | 2 | DHP, AMPQ | 116 | 63.8% | Any age; (20.6± 12.4) | Yes | Yes | People's Army Department of Military Medicine |
| 11 | Thanh, 2012 [32] | 5/2008–12/2009, | Vietnam | Commune centre | 2 | ASAQ, DHP | 128 | 70.1% | Any age; (18.9± 12.7) | | | Vietnam People's Army Department of Military Medicine |
| 12 | Thapa, 2007 [33] | 8/2005–10/2005. | Nepal | H (in-pt) | 2 | AL, SP | 99 | 53% (AL); 73% (SP) | >5 yr; (26.5 ± 13.8) | Yes | Yes | Not reported |
| 13 | Tjitra, 2001 [34] | 2007–2008 | Indonesia | 4 Hs | 2 | ASSP, SP | 105 | 60% | 83.8% (under 12 yr) | Yes | Yes | Nicholson-Hill Malaria Research Fund & Tudor Foundation. |
| 14 | Trung, 2009 [35] | NA | Vietnam, | treatment center (in-pt) | 2 | DHP, AMPQ | 103 | 61.2% | 25.8±13.9 | Yes | Yes | Science and Technology Research Projects of Guangdong Province |

(*Continued*)

**Table 1.** (Continued)

| | Author, Year of publication [ref.no] | Study period | Country | Setting | Study arms | Interventions | Participants | Male | Age, mean in year (±SD) or median and range | PCR | Under supervision | Funding |
|---|---|---|---|---|---|---|---|---|---|---|---|---|
| 15 | Valecha, 2010 [36] | NA | Multi country (India, Laos, Thailand) | OPD | 2 | DHP, ASMQ | 1150 | 78.4% | mainly adults, (25.46± 13.3) | Yes | Yes | MMV, Sigma Tau. & Oxford University |
| 16 | van Vgt,2000 [37] | 11/ 1997-3/1998 | Thailand | H & health camp | 2 | AL, ASMQ | 200 | 73.5% | Adults & children (23, 13–63) | | | Wellcome Trust of Great Britain |
| 17 | Wilairatana,2002 [38] | ?? | Thailand | H (in-pt) | 2 | DHP, ASMQ | 352 | 66.8% | 24.8 (±13.3) | No | Yes | Tonghe Phramaceutical Co. Ltd |

AL: Artemether-lumefantrine; AMPQ; artemisinin-piperaquine; ASMQ: artesunate-mefloquine; ASMQh: artesunate-mefloquine home treatment/not supervised; ASMQ2: artesunate-mefloquine 2-day course; ASAQ; artesunate-amodiaquine; ASCQ: artesunate-chloroquine; ASSP: artesunate plus sulfadoxine-pyrimethamine; CQ: chloroquine; DHP: dihydroartemisinin-piperaquine; DHP4: dihydroartemisinin-piperaquine 4-day course; DHPh; dihydroartemisinin-piperaquine home treatment/ not supervised; DHPAS dihydroartemisinin-piperaquine & artesunate added; SP: sulfadoxine-pyrimethamine; gr: group(s); H; hospital; In-pt: Inpatients; MMV: Medicines for Malaria Venture; MOST/China: Ministry of Science and Technology of the People's Republic of China; MSF: Medecins Sans Frontieres; OPD: outpatient department/centre; WHO/RBM: World Health Organization/Roll Back Malaria' yr: year.

## Discussion

### Summary of main results

The present network meta-analysis, including 14 different antimalarial interventions from 17 RCTs studies, provided both direct and indirect evidences regarding the relative efficacy at the end of 28-day follow-up time. This approach provided both direct and indirect information through the use of a common comparator to obtain estimates of the relative effects on multiple-intervention comparisons. To our knowledge, this is the first comprehensive synthesis of data from available antimalarial interventions for treatment of uncomplicated *falciparum* malaria in Asia.

If parasite populations are not completely eradicated from the patient, low-level replication continues until it reaches the microscopic or clinical detection threshold to cause a treatment failure (i.e. recrudescent infection). The parasitological cure rates at day 28 is potentially a more sensitive marker for *in vivo* efficacy of an antimalarial drug compared to the cure rates at day 14. This is particularly for drugs with long half-lives in plasma. But, it is also applicable to a certain me extent to drugs with short plasma half-lives. Therefore, the end point of PCR-corrected cure rate at day 28 in the current analysis is an acceptable outcome measure to increase the sensitivity of the test as well as to generate appropriate data for a comparison of the different treatment regimens (artesunate alone versus the combination of artesunate and AMQ) [41]. Overall, the results of this NMA provided low quality evidence that no regimen could provide better rates of treatment success, except DHP.

Artemisinin resistance to parasites have shown reduced susceptibility and is clearly associated with increasing rate of failure of ACT in Cambodia [42] and Thailand [43]. K13-propeller mutation was identified as a key determinant of artemisinin resistance in Southeast Asia [44]. Many administrative regions in Myanmar had combined K13-propeller mutation prevalence of more than 20% [45], including regions on the Myanmar India border areas.

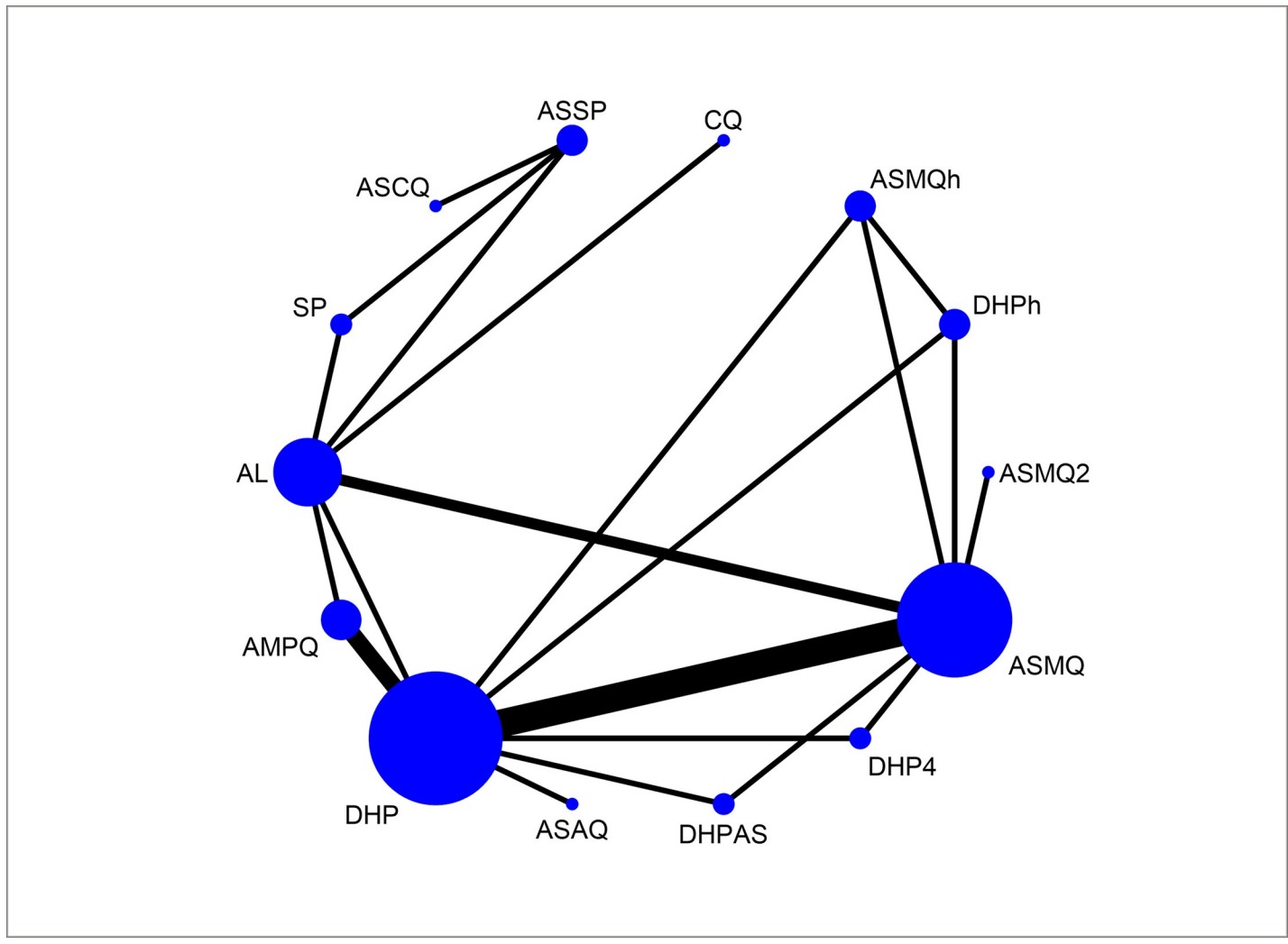

**Fig 2. Network plot of the antimalarials for treating *P. falciparum* malaria.**

ACT is recommended to be given for three days along with slowly eliminated partner drug. In a three-day regimen, the artemisinin component stays in body for two asexual parasite life-cycles, except for *P. malariae*. In each asexual cycle, artemisinin and its derivatives reduce parasite counts by a factor of almost 10,000 [47,48]. The current NMA showed that there is a very low-certainty evidence that DHP (3-day course) was superior to other ACTs (ASAP, AL, ASAQ, ASMQ). A published NMA of antimalarial treatments for uncomplicated *falciparum* malaria in African children that included 12 RCTs showed superiority of DHP among currently WHO recommended ACTs [46], but it did not report an overall quality of evidence. The utility of DHP is an important information in drug compliances as AL has to be administered twice daily for three days and it also need to take fatty food to be effective. Moreover, the treatment cost of AL is also more expensive (>10 US$ per treatment course) [46].

## Study limitations

Studies in other languages may have been missed, if abstracts in English language are not available. Future study, addressing the cost-effectiveness of particular antimalarial drug

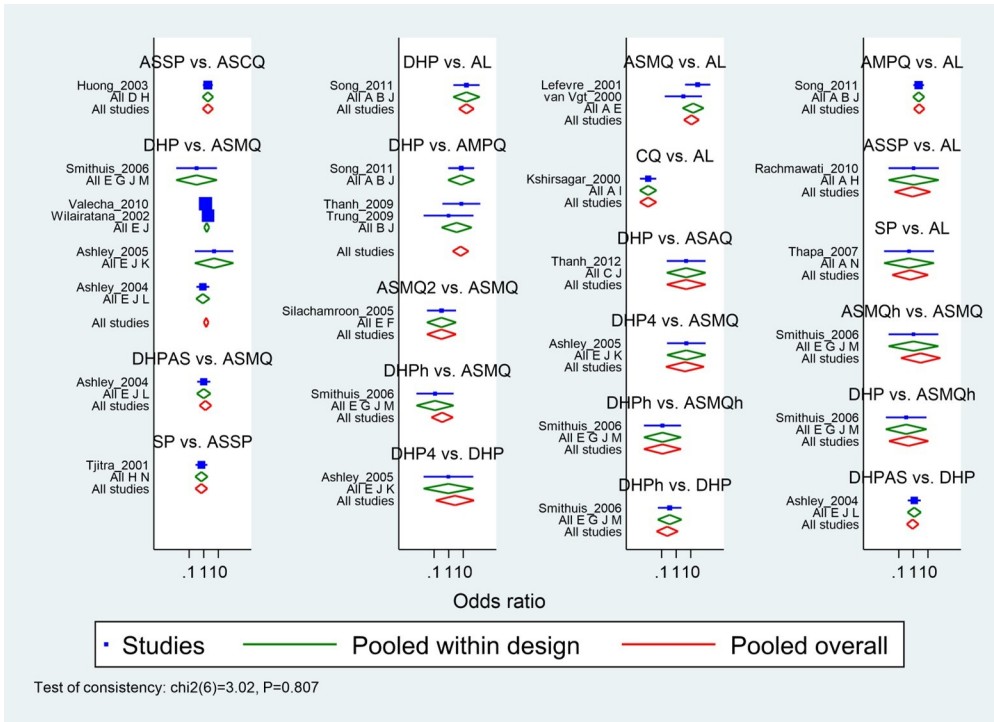

**Fig 3. All direct and mixed comparisons.**

**Table 2. GRADE quality of evidence for the comparative efficacy of antimalarial.**

| Treatment comparison | Direct estimate; OR (95% CI) | Quality of evidence | Indirect estimate; OR (95% CI) | Quality of evidence | Network estimate; OR (95% CI) | Quality of evidence |
|---|---|---|---|---|---|---|
| DHP vs AL | 1.29 (0.74 to 2.23) 1 fewer per 1,000 (from 2 fewer to 1 fewer) | ⊕○○○d,e VERY LOW | 2.5 (1.08 to 5.8) 3 fewer per 1,000 (from 6 fewer to 1 fewer) | ⊕⊕○○a LOW | 2.5 (1.08 to 5.8) 3 fewer per 1,000 (from 6 fewer to 1 fewer) | ⊕⊕○○a LOW |
| DHP vs ASCQ | NA | NA | 11.21 (3.40 to 36.89) | ⊕○○○ VERY LOW | 11.21 (3.4–36.89) 11 fewer per 1,000 (from 37 fewer to 3 fewer) | ⊕○○○a,b,c VERY LOW |
| DHP vs CQ | NA | NA | 16.54 (5.02–54.56) | ⊕⊕○○ LOW | 16.54 (5.02 to 24.56) 17 fewer per 1,000 (from 25 fewer to 5 fewer | ⊕⊕○○a,b LOW |
| DHP vs ASSP | NA | NA | 0.01 (0.00–0.04) | ⊕⊕○○ LOW | 0.01 (0.00 to 0.04) 0 fewer per 1,000 (from—to 0 fewer) | ⊕⊕○○a,b LOW |
| ASCQ vs AL | NA | NA | 0.22 (0.06–0.79) | ⊕⊕○○ LOW | 0.22 (0.06–0.79) 0 fewer per 1,000 (from 1 fewer to 0 fewer) | ⊕⊕○○a LOW |
| ASAQ vs AL | NA | NA | 5.55 (0.26–119.75) | ⊕○○○ VERY LOW | 5.55 (0.26–119.75) 6 fewer per 1,000 (from 120 fewer to 0 fewer) | ⊕○○○a,b,d VERY LOW |

CI: Confidence interval; OR: Odds ratio; Explanations: a. studies at unclear and high risk of bias; b. wide predictive interval; c. very wide CI; d. wide 95%CI and it crossed a null value; e: a singular study at high risk of bias; f. a singular study

00_-

| _SP | DHPh | DHPAS | DHP4 | DHP | CQ | ASSP | ASMQh | ASMQ2 | ASMQ | ASCQ | ASAQ | AMPQ | AL |
|---|---|---|---|---|---|---|---|---|---|---|---|---|---|
| SP | 74.73 (1.01, 5511.31) | 6.70 (0.20, 223.55) | 24.51 (0.99, 604.88) | 1.66 (0.10, 28.35) | 27.49 (1.27, 596.58) | 0.02 (0.00, 0.45) | 58.47 (0.74, 4640.46) | 1.45 (0.56, 3.73) | 6.00 (0.13, 279.95) | 18.62 (0.86, 403.85) | 0.75 (0.21, 2.62) | 5.70 (0.07, 438.98) | 4.15 (0.22, 80.09) |
| | DHPh | 0.09 (0.00, 2.84) | 0.33 (0.01, 7.63) | 0.02 (0.00, 0.56) | 0.37 (0.02, 7.56) | 0.00 (0.00, 0.01) | 0.78 (0.01, 59.62) | 0.02 (0.00, 1.43) | 0.08 (0.00, 3.55) | 0.25 (0.01, 5.05) | 0.01 (0.00, 0.80) | 0.08 (0.00, 5.65) | 0.06 (0.00, 1.45) |
| | | DHPAS | 3.66 (0.53, 25.08) | 0.25 (0.03,1.95) | 4.10 (0.76, 22.28) | 0.00 (0.00, 0.03) | 8.73 (0.46, 164.84) | 0.22 (0.01, 7.22) | 0.89 (0.05, 15.79) | 2.78 (0.50, 15.41) | 0.11 (0.00, 4.11) | 0.85 (0.03, 28.23) | 0.62 (0.08, 5.07) |
| | | | DHP4 | 0.07 (0.02, 0.3) | 1.12 (0.44, 2.85) | 0.00 (0.00, 0.01) | 2.39 (0.09, 61.28) | 0.06 (0.00, 1.46) | 0.24 (0.02, 2.94) | 0.76 (0.30, 1.94) | 0.03 (0.00, 0.84) | 0.23 (0.01, 5.74) | 0.17 (0.04, 0.80) |
| | | | | DHP [1st, 99.2%] | 16.54 (5.02, 54.56) | 0.01 (0.00, 0.04) | 35.19 (1.26, 982.69) | 0.87 (0.05, 14.88) | 3.61 (0.27, 48.26) | 11.21 (3.40, 36.89) | 0.45 (0.02, 8.65) | 3.43 (0.13, 92.10) | 2.50 (1.08, 5.8) |
| | | | | | CQ | 0.00 (0.00, 0.00) | 2.13 (0.09, 47.85) | 0.05 (0.00, 1.14) | 0.22 (0.02, 2.25) | 0.68 (0.47, 0.98) | 0.03 (0.00, 0.66) | 0.21 (0.01, 4.45) | 0.15 (0.04, 0.53) |
| | | | | | | ASSP | 2957.64 (82.65, 105837.4) | 73.26 (3.22, 1667.03) | 303.29 (16.61, 5539.00) | 941.85 (160.45, 5528.87) | 37.88 (1.50, 959.08) | 288.10 (8.34, 9946.47) | 210.14 (44.33, 996.16) |
| | | | | | | | ASMQh | 0.02 (0.00,1.97) | 0.10 (0.00, 4.98) | 0.32 (0.01, 7.25) | 0.01 (0.00, 1.10) | 0.10 (0.00, 7.70) | 0.07 (0.00, 2.03) |
| | | | | | | | | ASMQ2 | 4.14 (0.09, 193.36) | 12.86 (0.59, 278.97) | 0.52 (0.23, 1.18) | 3.93 (0.05, 303.24) | 2.87 (0.15, 55.33) |
| | | | | | | | | | ASMQ | 3.11 (0.31, 31.08) | 0.12 (0.00, 6.37) | 0.95 (0.02, 44.78) | 0.69 (0.05, 9.57) |
| | | | | | | | | | | ASCQ [2nd, 75.4%] | 0.04 (0.00, 0.97) | 0.31 (0.01, 6.71) | 0.22 (0.06, 0.79) |
| | | | | | | | | | | | ASAQ | 7.61 (0.09, 633.69) | 5.55 (0.26, 119.75) |
| | | | | | | | | | | | | AMPQ | 0.73 (0.03, 20.06) |
| | | | | | | | | | | | | | AL [3rd, 73.5%] |

**Fig 4. Network meta-analysis of antimalarial treatments.**

intervention is needed. Some RCTs included were done with small sample size and they also had low methodological quality. Hence, there was a concern to the confidence in the effect estimates. Due to lack of blinding in some RCTs included, the risk of performance bias is a concern. Myanmar has substantially more malaria than any other country in the Southeast Asia [45]. The current analysis included only one RCT from Myanmar, indicating a limited geographical representativeness and an interpretation of the findings was limited with regard to generalizability.

## Implications

There is evidence that DHP has better treatment outcome than monotherapy such as SP and CQ, which are almost completely ineffective in Southeast Asia due to drug resistance.

Evidence from RCTs may not apply to real-world practice, where people in need of antimalarial treatment are often limited to compliance due to unsupervised treatment and lack of monitoring. Further studies should be directed to detect the effectiveness in real-world practice and should also focus on monitoring and treatment regimens. Moreover, in the light of benefit versus harm concept, a NMA of the relative safety of different antimalarial treatment is needed.

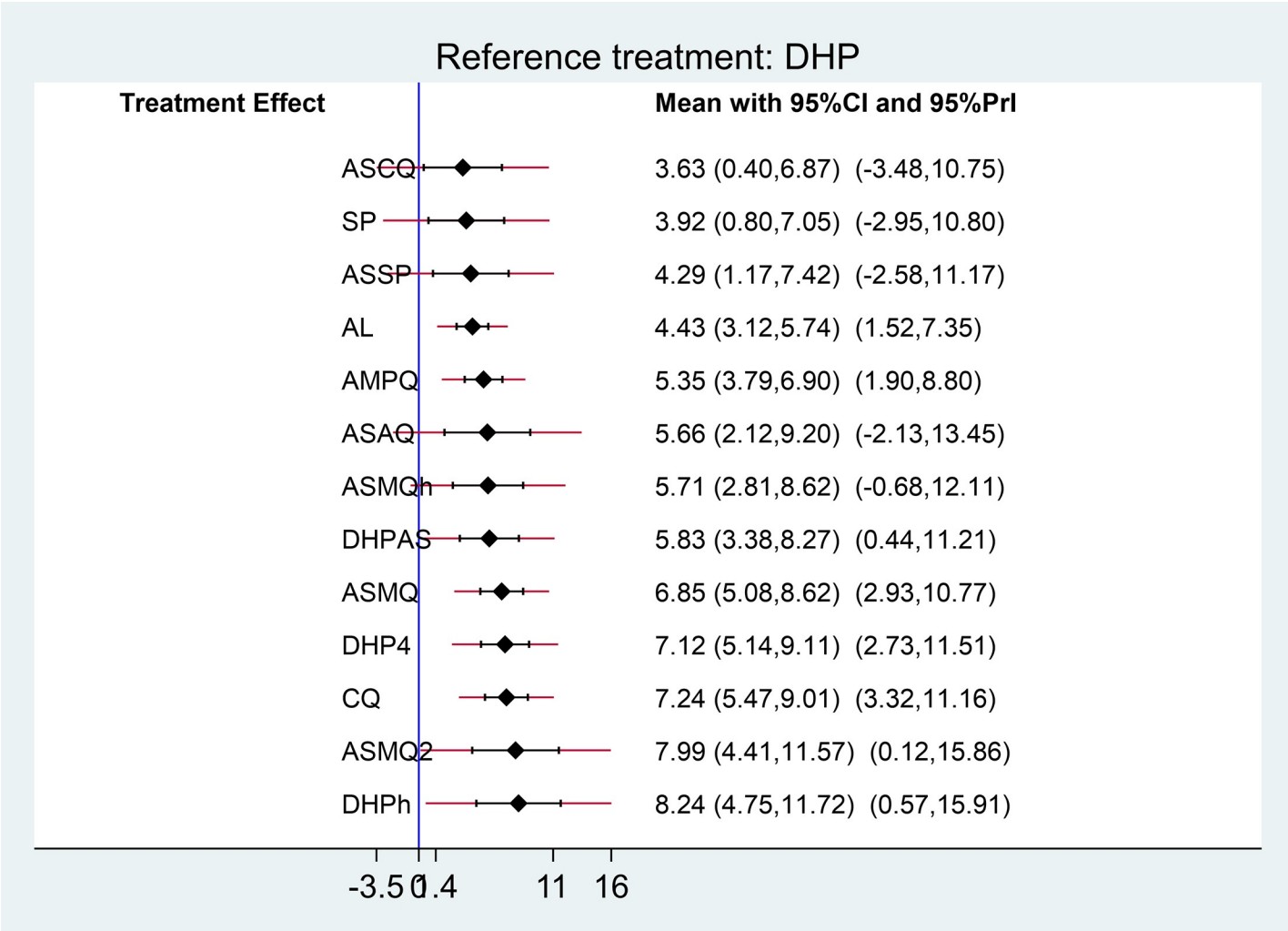

**Fig 5. Predictive intervals plot for the antimalarial network.**

## Conclusions

The findings suggest the superiority of DHP to AL and other comparator ACTs are with the overall low/very low quality of evidence judgements. Moreover, one drug regimen is better than another is only, if current drug-resistance patterns are at play. For example, the AL might be better than DHP in areas where both artemisinin and piperaquine resistance patterns are prevalent [7,9]. For substantiation, well-designed larger trials from endemic countries are needed. In the light of benefit versus harm concept, future analysis with safety information is recommended.

## Supporting information

**S1 Table. PRISMA NMA checklist.**
(PDF)

**S2 Table. Search terms.**
(PDF)

**S3 Table. Excluded studies and reasons for exclusion.**
(PDF)

**S4 Table. Distribution of studies and comparisons.**
(PDF)

**S5 Table. Risk of bias assessment by the review authors.**
(PDF)

**S6 Table. Treatment relative ranking.**
(PDF)

**S1 Fig. Forest plot of direct pairwise comparison of antimalarial regimens.**
(PDF)

**S2 Fig. Inconsistency plot with loop-specific heterogeneity.**
(PDF)

**S3 Fig. Comparison-adjusted funnel plot of the placebo-controlled antimalarial trials.**
(PDF)

**S4 Fig. Cumulative probability curves for the antimalarial network.**
(PDF)

## Acknowledgments

The authors thank the patients and researchers of the primary studies. We also thank our institutions for allowing us to perform this study. We also thank the anonymous reviewers and editors for giving us the comments and valuable inputs to improve the quality of our manuscript.

## Author Contributions

**Conceptualization:** Maxine A. Whittaker, Joon Wah Mak.

**Data curation:** Cho Naing, Norah Htet Htet, Saint Nway Aye.

**Formal analysis:** Cho Naing, Norah Htet Htet, Saint Nway Aye.

**Funding acquisition:** Joon Wah Mak.

**Investigation:** Cho Naing.

**Methodology:** Cho Naing, Maxine A. Whittaker, Norah Htet Htet, Joon Wah Mak.

**Project administration:** Cho Naing, Joon Wah Mak.

**Resources:** Joon Wah Mak.

**Supervision:** Joon Wah Mak.

**Writing – original draft:** Cho Naing, Maxine A. Whittaker.

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
