## [Decision Letter · Decision Letter 0]

5 Nov 2019

PONE-D-19-26846

Efficacy of antimalarial drugs for treatment of uncomplicated falciparum malaria in Asian region: A network meta-analysis

PLOS ONE

Dear Cho Naing,

Thank you for submitting your manuscript to PLOS ONE. After careful consideration, we feel that it has merit but does not fully meet PLOS ONE’s publication criteria as it currently stands. Therefore, we invite you to submit a revised version of the manuscript that addresses the points raised during the review process.

I would like to apologise for the time taken to come to a decision on your submission. This was due to the many reviewers contacted refusing, or simply not replying, when asked to examine your manuscript. Finally, one expert accepted and his recommendation is minor revision. If you clearly point out in your rebuttal letter how you modified your revision to address each and every one of the comments then I should be able to make a rapid editorial decision without sending your revision back out for review.

We would appreciate receiving your revised manuscript by Dec 20 2019 11:59PM. To enhance the reproducibility of your results, we recommend that if applicable you deposit your laboratory protocols in protocols.io, where a protocol can be assigned its own identifier (DOI) such that it can be cited independently in the future. For instructions see: http://journals.plos.org/plosone/s/submission-guidelines#loc-laboratory-protocols

We look forward to receiving your revised manuscript.

Kind regards,

Gordon Langsley

Academic Editor

PLOS ONE

Journal Requirements:

https://apps.who.int/iris/bitstream/handle/10665/275867/9789241565653-eng.pdf?ua=1

https://malariajournal.biomedcentral.com/articles/10.1186/1475-2875-11-153

http://www.ajtmh.org/content/journals/10.4269/ajtmh.2006.74.991#html_fulltext

https://bmcinfectdis.biomedcentral.com/articles/10.1186/s12879-018-3506-x

https://aac.asm.org/content/aac/47/3/901.full.pdf

In your revision ensure you cite all your sources (including your own works), and quote or rephrase any duplicated text outside the methods section. Further consideration is dependent on these concerns being addressed.

3.Thank you for stating the following financial disclosure:

"none".

Please provide an amended Funding Statement that declares *all* the funding or sources of support received during this specific study (whether external or internal to your organization) as detailed online in our guide for authors at http://journals.plos.org/plosone/s/submit-now.  

Please state what role the funders took in the study.  If any authors received a salary from any of your funders, please state which authors and which funder. If the funders had no role, please state: "The funders had no role in study design, data collection and analysis, decision to publish, or preparation of the manuscript."

4.  Please include your tables as part of your main manuscript and remove the individual files. Please note that supplementary tables (should remain/ be uploaded) as separate "supporting information" files.

Reviewers' comments:

Reviewer's Responses to Questions

**Comments to the Author**

1. Is the manuscript technically sound, and do the data support the conclusions?

Reviewer #1: Yes

2. Has the statistical analysis been performed appropriately and rigorously? 

Reviewer #1: Yes

3. Have the authors made all data underlying the findings in their manuscript fully available?

Reviewer #1: Yes

4. Is the manuscript presented in an intelligible fashion and written in standard English?

Reviewer #1: Yes

5. Review Comments to the Author

Reviewer #1: Suggest defining “Asian region” in Methods-Study selection (since this definition was used to include studies in the analysis) or the Introduction (since line 84 may suggest to some readers that Asian region is synonymous with GMS).

Suggest defining uncomplicated malaria and indicating whether this definition includes patients with hyperparasitemia but without signs or symptoms of severe malaria.

Line 45: Suggest assigning abbreviations for all treatments at first use here and using them consistently throughout the manuscript, tables, and figures.

Line 57: The conclusion that DHP is superior to AL is not supported by the OR (95% CI) data in line 57 and Table 2.

Lines 59, 308, 312: The suggestion to analyze safety information in future studies seems to come out of nowhere. Does it relate to the benefit vs. harm concept mentioned in line 242? In any case, suggest mentioning at least one result in the Abstract to support this statement.

Line 100: Suggest providing 1-2 examples of conflicting results in other studies that the present study was designed to address.

Line 108: This sentence suggests that ASCQ (mentioned in the Abstract) is currently used in Asian region. Is this true? If so, where? How is “currently” defined?

Line 154: Does “preferred” mean that the ITT population analysis was used over another analysis whenever it was available, or that only ITT population analyses were used?

Line 191: Suggest indicating that 5043 refers to total number of patients.

Line 197: Suggest stating the range of % males as well.

Line 302: This statement seems misleading considering that SP and CQ are almost completely ineffective in Southeast Asia due to drug resistance. Suggest revising.

In general, the authors should state the caveat that one drug regimen is better than another only if current drug-resistance patterns are at play. For example, AL may be far superior to DHP in areas where both artemisinin and piperaquine resistance are prevalent.

Table 1: Funding for Studies 1, 12 and 16 are missing. If information is not available, suggest stating so.

6. PLOS authors have the option to publish the peer review history of their article (what does this mean?). If published, this will include your full peer review and any attached files.

Reviewer #1: No

---

## [Author Response · Author response to Decision Letter 0]

13 Nov 2019

Q/1 Suggest defining “Asian region” in Methods-Study selection 

(since this definition was used to include studies in the analysis) 

or the Introduction (since line 84 may suggest to some 

readers that Asian region is synonymous with GMS). 

A/1 We have provided an operational definition of the Asian region.Thank you 

Methods

Selection of study

 An operational definition of the Asian region for this particular study covers countries in three regions of Southeast Asia, South Asia and East Asia.

/2 Suggest defining uncomplicated malaria and indicating whether this definition includes patients with hyperparasitemia but without signs or symptoms of severe malaria. 

A/2

According to the WHO revised criteria, P. falciparum parasitaemia >5% or > 200,000/μl is one of the feature of severe malaria.

In the present meta-analysis study, some studies indicated specific information on parasitemia as less than 200,000/μl. But, not all trials provided such information. All studies had confirmed P. falciparum and excluded patients with clinical features of severe malaria as described by WHO. 

Hence, we selected the studies if they had indicated patients with uncomplicated malaria (excluding severe malaria) are recruited.

As suggested, we now have provided additional information for clarification. Thank you 

Selection of study 

Uncomplicated malaria caused by the P. falciparum parasite in this study is defined as patients with the presence of P. falciparum and having symptoms that are non-specific and with no clinical or laboratory findings of severe organ dysfunction.

Q/3 Line 45: Suggest assigning abbreviations for all treatments at first use here and using them consistently throughout the manuscript, tables, and figures. 

A/3 We have updated the abbreviations in consistency manner throughout the manuscript. In the abstract, please allow us to use abbreviations, only if it is necessary to repeat. Thank you

Q/4 Line 57: The conclusion that DHP is superior to AL is not supported by the OR (95% CI) data in line 57 and Table 2. 

A/4 Please, accept our apology for a typing mistake in the previous version. It should be described as (OR 2.5, 95%CI: 1.08 to 5.8) as shown in Table 2 pertinent to ‘Network estimate’ column. We have done the corrections in this revised version

Moreover, we have briefly added a caution over drug resistance status of the area. Thank you for giving the valuable inputs. 

Abstract 

Results

In general, DHP had better efficacy than AL at day 28 (DHP vs AL: OR 2.5, 95%CI:1.08-5.8). There is low certainty evidence due to limited number of studies and small trials. 

 Conclusion

The findings suggest the superiority of DHP (3–day course) to AL and other comparator ACTs are with the overall low/very low quality of evidence judgements.

 Moreover, one drug regimen is better than another is only, if current drug-resistance patterns are at play. For example, the AL might be better than DHP in areas where both artemisinin and piperaquine resistance patterns are prevalent.

Text

Results

Fourteen-node analysis

For instance, DHP compared to AL in a single trial and there was a superiority of DHP in cure rate at day 28 (OR 2.5, 95%CI: 1.08 to 5.8) (Table 2)

Q/5 Lines 59, 308, 312: The suggestion to analyze safety information in future studies seems to come out of nowhere. Does it relate to the benefit vs. harm concept mentioned in line 242? In any case, suggest mentioning at least one result in the Abstract to support this statement. 

Q/5 We gratefully agreed on the inputs given. 

We have recommended future studies to included safety analysis so as to cover the benefit versus harm concept.

Thank you 

A/5 Abstract

Conclusions

In the light of benefit versus harm concept, future analysis including safety information are recommended.

Discussion

Implications

Moreover, in the light of benefit versus harm concept, a NMA of the relative safety of different antimalarial treatment is needed.

Conclusions

In the light of benefit versus harm concept, future analysis with safety information is recommended.

Q/6 Line 100: Suggest providing 1-2 examples of conflicting results in other studies that the present study was designed to address. 

A/6 We have added salient examples in brief. Thank you 

Text

Introduction

For example, a review by Zani and associates reported that in Africa, there was better efficacy in DHP than AL at day 28. However, such relationship was not shown in Asia [7]. This reflects that efficacy of antimalarial is related to whether the area is with artemisinin-sensitive parasite populations or not [6,9]. 

Q/7 Line 108: This sentence suggests that ASCQ (mentioned in the Abstract) is currently used in Asian region. Is this true? If so, where? How is “currently” defined? 

A/7 Our main focus was the currently used 5 ACTs including DHP, AL. These currently used drugs compared with other antimalarials. All comparators are not currently used antimalarials (e.g ASCQ in this case). 

We have indicated the comparator drugs as available in the trials. 

For clarity, we have removed this result from abstract. 

To be more specific, we have added a concern over an issue of drug resistance for an interpretation of the results. We very much appreciated the points provided. Thank you 

Abstract

Results

In general, DHP had better efficacy than AL at day 28 (DHP vs AL: OR 2.5, 95%CI:1.08-5.8).There is low certainty evidence due to limited number of studies and small trials. 

Conclusions

The findings suggest the superiority of DHP (3–day course) to AL and other comparator ACTs are with the overall low/very low quality of evidence judgements. Moreover, one drug regimen is better than another is only, if current drug-resistance patterns are at play. For example, the AL might be better than DHP in areas where both artemisinin and piperaquine resistance patterns are prevalent.

Q/8 Line 154: Does “preferred” mean that the ITT population analysis was used over another analysis whenever it was available, or that only ITT population analyses were used? 

A/8 For clarity, we have updated this information. Thank you for the valuable input. 

Data synthesis

We preferred the intention-to-treat (ITT) analysis over another analysis whenever it was available.

Q/9 Line 191: Suggest indicating that 5043 refers to total number of patients.We have clarified this information.

 Thank you 

A/9 Results

After removal of duplicates and screening of titles and abstracts, 34 full-text papers were evaluated and 17 studies, incorporating 5043 total number of patients were finally included in this review [14, 23-38].

Q/10 Line 197: Suggest stating the range of % males as well. 

A/10 We have updated this information. Thank you The majority of participants in the studies included were males (range from 51% to 96%) with mean age between 5.9 to 29 years. 

Q/11 Line 302: This statement seems misleading considering that SP and CQ are almost completely ineffective in Southeast Asia due to drug resistance. Suggest revising. We have clarified the statement. Thank you 

Implications

 There is evidence that DHP has better treatment outcome than monotherapy such as SP and CQ, which are almost completely ineffective in Southeast Asia due to drug resistance.

Q/12 In general, the authors should state the caveat that one drug regimen is better than another only if current drug-resistance patterns are at play. For example, AL may be far superior to DHP in areas where both artemisinin and piperaquine resistance are prevalent. Much appreciated for giving the valuable inputs.

A/12 We have updated the conclusion, taking the points provided. Thanking you 

Abstract

Conclusions

 Moreover, one drug regimen is better than another is only if current drug-resistance patterns are at play. For example, the AL might be better than DHP in areas where both artemisinin and piperaquine resistance patterns are prevalent. 

Text

Conclusions

 The findings suggest the superiority of DHP to AL and other comparator ACTs are with the overall low/very low quality of evidence judgements. Moreover, one drug regimen is better than another is only, if current drug-resistance patterns are at play. For example, the AL might be better than DHP in areas where both artemisinin and piperaquine resistance patterns are prevalent [7,9]. 

Q/13 Table 1: Funding for Studies 1, 12 and 16 are missing. If information is not available, suggest stating so. 

A/13 Thank you for pointing out the missing information. We have updated the Table 1 accordingly.

 Table 1 

1 When submitting your revision, we need you to address these additional requirements.

http://www.journals.plos.org/plosone/s/file?id=wjVg/PLOSOne_formatting_sample_main_body.pdf and http://www.journals.plos.org/plosone/s/file?id=ba62/PLOSOne_formatting_sample_title_authors_affiliations.pd

A/ We have followed the PLOS One format.

Thank you

2 We noticed you have some minor occurrence of overlapping text with the following previous publication(s), which needs to be addressed:

https://apps.who.int/iris/bitstream/handle/10665/275867/9789241565653-eng.pdf?ua=1

https://malariajournal.biomedcentral.com/articles/10.1186/1475-2875-11-153

http://www.ajtmh.org/content/journals/10.4269/ajtmh.2006.74.991#html_fulltext

https://bmcinfectdis.biomedcentral.com/articles/10.1186/s12879-018-3506-x

https://aac.asm.org/content/aac/47/3/901.full.pdf

In your revision ensure you cite all your sources (including your own works), and quote or rephrase any duplicated text outside the methods section. Further consideration is dependent on these concerns being addressed. We have updated the whole text and improved any duplication.

A/ We have checked % similarity and it shows 20% similarity (attached as other file). We sincerely feel that this is an acceptable limit for a meta- analysis study. Thank you

3 Thank you for stating the following financial disclosure: “none".

a. Please provide an amended Funding Statement that declares *all* the funding or sources of support received during this specific study (whether external or internal to your organization) as detailed online in our guide for authors at http://journals.plos.org/plosone/s/submit-now. 

b. Please state what role the funders took in the study. If any authors received a salary from any of your funders, please state which authors and which funder. If the funders had no role, please state: "The funders had no role in study design, data collection and analysis, decision to publish, or preparation of the manuscript."

4 Please include your tables as part of your main manuscript and remove the individual files. Please note that supplementary tables (should remain/ be uploaded) as separate "supporting information" files 

A/We have placed Tables in the manuscript. Thank you

---

## [Decision Letter · Decision Letter 1]

15 Nov 2019

Efficacy of antimalarial drugs for treatment of uncomplicated falciparum malaria in Asian region: A network meta-analysis

PONE-D-19-26846R1

Dear Dr. Cho Naing,

We are pleased to inform you that your manuscript has been judged scientifically suitable for publication and will be formally accepted for publication once it complies with all outstanding technical requirements.

With kind regards,

Gordon Langsley

Academic Editor

PLOS ONE

Additional Editor Comments (optional):

Reviewers' comments:

Reviewer's Responses to Questions

**Comments to the Author**

1. If the authors have adequately addressed your comments raised in a previous round of review and you feel that this manuscript is now acceptable for publication, you may indicate that here to bypass the “Comments to the Author” section, enter your conflict of interest statement in the “Confidential to Editor” section, and submit your "Accept" recommendation.

Reviewer #1: All comments have been addressed

2. Is the manuscript technically sound, and do the data support the conclusions?

Reviewer #1: Yes

3. Has the statistical analysis been performed appropriately and rigorously? 

Reviewer #1: Yes

4. Have the authors made all data underlying the findings in their manuscript fully available?

Reviewer #1: Yes

5. Is the manuscript presented in an intelligible fashion and written in standard English?

Reviewer #1: Yes

6. Review Comments to the Author

Reviewer #1: Authors: Thank you for addressing my comments and incorporating revisions into the manuscript, much appreciated.

7. PLOS authors have the option to publish the peer review history of their article (what does this mean?). If published, this will include your full peer review and any attached files.

Reviewer #1: No

---

## [Editor Report · Acceptance letter]

21 Nov 2019

PONE-D-19-26846R1 

Efficacy of antimalarial drugs for treatment of uncomplicated falciparum malaria in Asian region: A network meta-analysis 

Dear Dr. Naing:

I am pleased to inform you that your manuscript has been deemed suitable for publication in PLOS ONE. Congratulations! Your manuscript is now with our production department. 

With kind regards,

on behalf of

Dr. Gordon Langsley 

Academic Editor

PLOS ONE